# What do blind people "see" with retinal prostheses? Observations and qualitative reports of epiretinal implant users

**Cordelia Erickson-Davis[1]⚬\*, Helma Korzybska[2]⚬\***

**1** Stanford School of Medicine and Stanford Anthropology Department, Stanford University, Palo Alto, California, United States of America, **2** Laboratory of Ethnology and Comparative Sociology (LESC), Paris Nanterre University, Nanterre, France

⚬ These authors contributed equally to this work.
\* cred22@stanford.edu (CED); helmakorzybska@gmail.com (HK)

**Data Availability Statement:** The datasets cannot be shared publicly, as they contain both identifying and sensitive participant information. The small

## Abstract

### Introduction

Retinal implants have now been approved and commercially available for certain clinical populations for over 5 years, with hundreds of individuals implanted, scores of them closely followed in research trials. Despite these numbers, however, few data are available that would help us answer basic questions regarding the nature and outcomes of artificial vision: what do recipients see when the device is turned on for the first time, and how does that change over time?

### Methods

Semi-structured interviews and observations were undertaken at two sites in France and the UK with 16 recipients who had received either the Argus II or IRIS II devices. Data were collected at various time points in the process that implant recipients went through in receiving and learning to use the device, including initial evaluation, implantation, initial activation and systems fitting, re-education and finally post-education. These data were supplemented with data from interviews conducted with vision rehabilitation specialists at the clinical sites and clinical researchers at the device manufacturers (Second Sight and Pixium Vision). Observational and interview data were transcribed, coded and analyzed using an approach guided by Interpretative Phenomenological Analysis (IPA).

### Results

Implant recipients described the perceptual experience produced by their epiretinal implants as fundamentally, qualitatively different than natural vision. All used terms that invoked electrical stimuli to describe the appearance of their percepts, yet the characteristics used to describe the percepts varied significantly between recipients. Artificial vision for these recipients was a highly specific, learned skill-set that combined particular bodily techniques, associative learning and deductive reasoning in order to build a "lexicon of flashes"—a distinct perceptual vocabulary that they then used to decompose, recompose and interpret their

number of participants implanted with these devices and the nature of the interview content (e.g. the focus on individual participant life histories and candid responses) make anonymization of the data unfeasible. Participants also gave their consent with the understanding that their responses would remain confidential. However, other researchers may send data access requests to the authors, the Stanford University institutional review board (irbeducation@stanford.edu), or the Paris Nanterre University review board (nawale. lamrini@parisnanterre.fr).

**Funding:** CED - NSF Award #1753701. National Science Foundation https://nsf.gov/awardsearch/ showAward?AWD_ID= 1753701&HistoricalAwards=false. The funders had no role in study design, data collection and analysis, decision to publish, or preparation of the manuscript. CED - Dissertation Fieldwork Grant, Wenner Gren Foundation http://www.wennergren. org/grantees/erickson-davis-cordelia-roberta. The funders had no role in study design, data collection and analysis, decision to publish, or preparation of the manuscript.

**Competing interests:** The authors have declared that no competing interests exist.

surroundings. The percept did not transform over time; rather, the recipient became better at interpreting the signals they received, using cognitive techniques. The process of using the device never ceased to be cognitively fatiguing, and did not come without risk or cost to the recipient. In exchange, recipients received hope and purpose through participation, as well as a new kind of sensory signal that may not have afforded practical or functional use in daily life but, for some, provided a kind of "contemplative perception" that recipients tailored to individualized activities.

## Conclusion

Attending to the qualitative reports of implant recipients regarding the experience of artificial vision provides valuable information not captured by extant clinical outcome measures.

## Introduction

Retinal prostheses are implantable microelectronic devices designed to replace the function of phototransducing cells within the eyes of individuals with retinal diseases such as retinitis pigmentosa. The devices capture light from a camera image and then transform and transmit those data in the form of electrical impulses to the remaining cells within the retina. In an ideal visual prosthesis, the electrical impulses would perfectly mimic the signals of the cells that have been lost, and the individual's visual perception would be restored to what they remember of natural vision.

Because it is difficult to replicate biological infrastructure with microelectrode arrays, the goal has been to create a simplified visual signal that would provide some functional benefit for the recipient. Efforts to develop such a device have been undertaken by many groups over the past 50 years [1–3]. More recent efforts have focused on the retina, with three devices having achieved commercial approval to date. The first to market was the Argus II epiretinal prosthesis (Second Sight Medical Products, Sylmar, CA) a 60-electrode device that has been implanted in approximately 300 individuals since its commercial approval in the EU in 2011, the US in 2013 and in Canada 2015 [4–7]. Retinal Implant AG (Reutlingen, Germany) followed soon thereafter with the Alpha IMS and AMS implants—two versions of a 1500 electrode subretinal device that was first approved in the EU in 2013 and that has been implanted in approximately 70 individuals [8, 9]. Finally, there was the IRIS II produced by Pixium Vision (Paris, France), a 150-electrode epiretinal implant that received approval in the EU in 2016 [10]. Meanwhile, there are many other retinal-based implants and alternative approaches for artificial vision in various stages of development [11–14].

The companies that have produced the commercially-approved retinal devices have claimed that the devices can "provide useful vision" to individuals severely impacted by RP (retinitis pigmentosa) by allowing recipients to "distinguish and interpret patterns (. . .) recognize outlines of people, basic shapes and movement (. . .) and navigate more independently through the world." [15]. However, the literature reporting clinical outcomes of these devices indicates that the reality is more complicated and ambiguous than these statements might convey [7–9, 16–23]. No device recipient has achieved projected acuity goals, and even the best performing recipients have not improved to the level of "legal blindness" on standard measures of acuity. Because of this, manufacturers and associated clinical groups have developed novel test methods for "ultra-low vision," including perception of light, light localization,

direction of motion, as well as "real world" functional tasks such as object localization and recognition, sock sorting, sidewalk tracking and walking direction tests [5, 8, 24]. However, it is difficult to make sense of reported outcomes using measures given the novelty of the tasks and lack of consistency between the groups utilizing them, making comparative analysis between groups and devices difficult.

Several guideline and consensus-building endeavors have been launched in recent years in order to address the issue of heterogeneity in outcomes [25–27]. Yet the question of which measures of visual function should be used to assess the outcomes of these early-stage devices remains an area of active debate. There is no agreement regarding the best way to report outcomes with these devices in part because it is unclear what kind of vision is being produced. It has thus become a reinforcing cycle: there is not a good idea what artificial vision is "like" because there are not good outcome measures, and there are not good outcome measures because it is unclear what artificial vision is "like."

Here, the authors assert that what is needed to move beyond this impasse is a more holistic, qualitative understanding of the perceptual experience associated with these devices. Extant research has overlooked the subjective accounts of users, both regarding the quality of the perceptual experience associated with these devices (i.e. what are the percepts generated by devices "like"), as well as the more general experience of what it is like to receive, learn to use and live with these devices. A handful of studies have included qualitative reports of elicited percepts as part of psychophysical tasks [8, 22, 28, 29], and certain companies have included recipient reports of objects they are able to discern in their daily life (e.g. Alpha IMS recipient reported being able to see a smile [21], but rarely has there been a substantive account of what artificial vision "is like" [30], nor what it is like to use one of these devices (i.e. in what form does that smile reported by the Alpha IMS recipient appear to them, and how did they use the device to achieve that?).

A greater understanding of the qualitative experience associated with retinal implants is indispensable to the development of visual prostheses and related technologies, as well as to our understanding of vision more generally. More complete accounts of recipients' perceptual experience can inform training and rehabilitation strategies, as well as aid in the development of a more accurate model of artificial vision. This will be of use to researchers and manufacturers developing these devices (e.g. comparing predictions of artificial vision to actual recipient reports to allow feedback necessary for successful iterative development of these devices), as well as to potential recipients and their families (a more accurate model of what artificial vision is like would better inform their decision regarding receiving a similar device). Finally, as far as subjective experience is the *sine qua non* of consciousness, subjective reports provide a kind of understanding that even the most detailed mechanistic accounts of the biological underpinnings of vision or complete battery of behavioral measures could ever capture.

Here we bring to bear the theoretical and methodological tools of cultural anthropology ethnography in order to address larger questions regarding perceptual experience of visual sensory prostheses. We build upon a growing body of literature in the social sciences that focuses on "disability" and ways of experiencing prosthetics (e.g. prosthetic arms, wheelchairs, cochlear implants) [31–34]. This body of work is founded on the assumption that the cultural environment of a recipient shapes her perceptual experience [35–42]. By using the tools of ethnography and attending to the beliefs and practices of our interlocutors—here recipients of visual prostheses and rehabilitation specialists—we offer an account of how the design and implementation of a complex medical technology translates into perceptual experience.

Ethnography is a form of qualitative research that provides rich, holistic descriptions of particular social phenomena through the collection of detailed observations and interviews. Rather than being hypothesis-driven, the emphasis in ethnography is on exploring the nature

of a phenomena using unstructured data and an inductive analytic approach in which themes or findings "emerge" from the data. Where quantitative analysis aim is to classify features, count and construct statistical models in an attempt to explain what is observed, the aim in qualitative studies is a complete, detailed description that can give insight to practices and experiences that are normally "hidden" from the public gaze. These emergent insights can be used as hypotheses in future studies, rendering qualitative and quantitative methods entirely complementary.

Here we present our findings regarding the perceptual experience associated with epiretinal implants in the context of getting, learning to use and live with the device. In addition to presenting novel insights regarding an important new genre of neurotechnology, the study serves as a case example of how perceptual data—and the phenomenological and ethnography methods utilized to capture them—can be used to think more about the processes and effects of emerging medical technologies. We will discuss the implications of these findings, and point out ways in which they can be used to direct future research.

## Methods

This research was conducted as a part of ethnographic fieldwork that both authors completed for their respective dissertation projects on the experience of retinal implant recipients. Both authors were involved in recruitment and data collection, and both were trained in qualitative research, research governance, ethics, and adhered to standard ethnographic procedures.

The study utilized an ethnographic approach in order to explore the experience of individuals receiving and working with epi-retinal prosthesis devices. The authors performed thematic analysis using data collected from: 1) individual, semi-structured interviews with implant recipients, visual rehabilitation staff and industry researchers, as well as of 2) field notes the authors took based on their observations of the interactions of those individuals at those sites. Ethnography is a methodological approach committed to providing in-depth accounts of everyday lived experience and practice. Ethnographic inquiry centers around recipient observation—when one lives and works with the communities they are studying, taking careful note of the daily interactions and practices. It also includes archival research as well as both formal and informal interviews. Qualitative research methods such as these allow researchers and recipients to discover and explore topic areas without predetermined questions. In-depth, semi-structured interviews allow for the collection of valuable information in diverse settings, while observation provides useful context and additional information that recipients may be unable or unwilling to share [43, 44].

This study was conducted in accordance with guidelines for the Consolidated Criteria for Reporting Qualitative Research (COREQ) [44, 45]. and was approved by the Stanford University institutional review board (protocol number 33528), the Paris Nanterre University review board and CNIL General Data Protection Regulation (GDPR) (declared under MR0017220719) as well as at the individuals at hospital sites. The ethics evaluation committee of Inserm, the Institutional Review Board (IRB00003888, IORG0003254, FWA00005831) of the French Institute of medical research and Health, has reviewed and approved this research project (HK).

### Participants

Purposive sampling sought to recruit individuals receiving implant devices ("recipients") over the two-year period of the study, as well as staff of varying experience of working with implant recipients at the two hospital sites (one in London, UK and one in Paris, France). Sampling

continued until sufficient data were obtained to address the research aims, which was determined by agreement of the authors.

Information about the ethnographic projects was presented by the authors to hospital staff at the respective sites. Permission for the authors to sit in to observe and speak to recipients was requested of the hospital administrators, the recipients as well as the vision rehabilitation specialists and company staff. Recipients recruited by author CED gave verbal consent, as was approved by the Stanford University IRB protocol (33528). Recipients recruited by HK gave verbal consent after being read consent forms they received, as approved by the French Institute of medical research and Health IRB protocol (19–593) and by the Paris Nanterre University review board. All recipients were given information about the research purpose, assurance that they could leave the study at anytime, as well as opportunity for contact information and review board reference. Additional permission and verbal consent was requested of the recipients before each observed session and interview.

All implant recipients who were asked over the two-year period agreed to speak with the authors and permitted them to observe their training. This resulted in a total of 16 implant recipients consisting 11 men and 5 women between the ages of 45 and 80 years. Thirteen of the recipients were based at a hospital in Paris and were French-speaking, and 3 of the implant recipients were based at a hospital in London, UK. These 16 recipients had received either Argus II or IRIS II epi-retinal devices. In addition, interview and observational data were collected with 8 vision rehabilitation specialists (four at the French hospital, four at the British hospital) as well as 8 industry researchers from Pixium Vision and Second Sight. Statements from nine of these device recipients will be presented in this article, whose reports we found to be representative of the entire consortium. Pseudonyms are used for all recipients to protect anonymity (e.g. "Recipient 1"). Transcripts from interviews with French-speaking recipients were translated from French to English by HK.

### Interview and observations

Using an exploratory descriptive qualitative methodology, individual, in-depth interviews were conducted with implant recipients, vision rehabilitation specialists and industry researchers from both companies at the two hospital sites. Open ended, semi-structured interviews were conducted and all recipients were interviewed by one of the two authors. Face-to-face interviews were conducted at an agreed-upon place, at the hospital during the days of their trainings. On occasion follow up questions were asked of recipients over the phone. These interviews were either tape -recorded upon patient agreement and subsequently transcribed, or notes were taken directly during informal conversations. A description of interview questions can be found in the supplementary material.

Interview data were then supplemented with observational data collected by the authors as they observed the training and rehabilitation processes within the hospitals. This allowed the researchers to examine not just what recipients said, but also what they did, and the setting or context in which their reports and practices took place. These observations were translated into field notes that together with the transcripts of the interviews made up the data set used in analysis.

### Data analysis

The interview and observational data were analyzed using Interpretative Phenomenological Analysis (IPA) [46]. This particular form of qualitative analysis was selected because of its emphasis on how recipients experience their world. It is useful for approaching research questions aimed to understand what an experience is like (phenomenology) and how someone

made sense of it (interpretation). Like other forms of ethnographic analysis, it is undertaken in an inductive thematic manner: themes or categories are derived from the data itself, rather than using predefined categories. Through a careful analysis of the data, using this inductive process, ethnographers generate tentative theoretical explanations from their empirical work [47].

This analysis identified key themes or findings in the experiences of prosthesis users, discussed below under the headings of 1) getting the device 2) learning to use the device and 3) living with the device. These headings follow the general chronology of receiving one of these devices, though individual findings that are discussed under each heading are drawn from analysis and consideration of all observational and interview data as a whole.

## Results

### Getting the device

**Election.**   The individuals we observed and spoke with—as well as the majority who have elected to receive these devices—were diagnosed with retinitis pigmentosa (RP), a group of disorders that involves the gradual degeneration of the eye's light-sensitive cells. These individuals had fully functioning visual systems before the first symptoms appeared—often in adulthood—and had functioned in the world as sighted persons until they were no longer able. Those who were older at disease onset often did not learn the compensatory or assistive techniques that are more readily available to younger people with visual impairments (e.g. schools for the blind, which teach braille and other techniques).

Individuals have unique life histories and reasons for wanting the device, but commonalities underlying most of their stories was the desire for greater independence and autonomy, to contribute to research for future generations (a hereditary disease, some of the individuals had children who had since been diagnosed), and a desire to challenge themselves. Whether it be an assistive tool with which to supplement their mobility, something that would allow them to return to the workforce or allow them greater social connection to individuals around them, the individuals we spoke with desired greater agency and connection within the world around them. We also found that recipients often expressed wanting to prove their capability (to themselves as well as others), as a kind of psychological emancipation from the "handicap" status they unwillingly represented.

According to industry researchers we spoke with, the most important predictor of device success is recipient selection; in particular the psychological profile of individual recipients who elect to get the device. The ideal recipients, these researchers said, were soldiers and former athletes. They had the endurance and commitment to "do what it took" to get through the training. They had a self-sacrificial mentality and often a stoicism that made them attractive recipients. These individuals were referred to by clinical researchers as "fighters" ("*des battants*"). "Excellent" candidates who were neither soldiers nor athletes but who were thought to have the qualities of a "fighter" included individuals who held jobs that involved challenging cognitive tasks (e.g. a computer scientist or a teacher). Other predictors of "successful" recipients included shorter disease duration and younger age. A "reasonable" recipient was someone who both met the diagnostic criteria and whom they judged to have "realistic" expectations. Tempering expectations, we would hear many of these researchers say, is a crucial factor in recipient selection and preparation. Recipients who had accepted or come to terms with their low vision condition often do best, a rehabilitation specialist stated. They are more likely to accept the difference between the reality of artificial vision to what they might have been expecting it to be like.

**Implantation.** Both of the epiretinal devices consist of an external (wearable) and internal (surgically implanted) component. The external equipment includes custom glasses that house a video microcamera connected by a wired cable to a processing unit. The processing unit transforms video from the camera to data that are then transmitted to the surgically placed internal implant. The latter is composed of a coil and electronics case, as well as either a 60 or 150-electrode array that is fixed to the inner surface of the retina [48]. Wireless communication from the external processing unit stimulates electrodes within the array to emit small electrical pulses that excite remaining viable inner retina cells. Unlike the wearable part of the device that can be cleaned or repaired, if there is a problem with the internal component, there is not much that can be done to adjust or repair, and will only be "explanted" if absolutely necessary.

In receiving one of these devices, recipients underwent a 2–6 hour surgery (depending on the device and the surgeon's experience) under general anesthesia [49]. It then takes approximately four weeks to heal from the surgery, during which most recipients reported that they did not mind the aftereffects of the surgery so much as their inability to go about their daily activities and sleep on their side [50]. Mostly we found the recipients were eager for the four weeks to be up and for the device to be activated.

**Initial activation.** A number of weeks after implantation, but before the camera is turned on, the device is aligned and activated in order to make sure the device fits correctly, that the base components are functioning, and to introduce the recipient to the perception invoked with electrical stimulation. They follow with the "systems fitting," in which global thresholds or settings for the stimulation parameters that yields a reliably "good" perception are determined (first amplitude followed by phase duration and frequency). The recipients' expectations are tempered once again at this point: they are told that this is not when they will "see" (they are told this will be when the camera is turned on). Nevertheless, we came to learn that this two-day period is associated with significant change and learning, as the recipient learns to identify the signal, or "phosphene"–the building block of artificial vision.

The devices are built and rehabilitation protocols implemented with the expectation that activation of a single electrode will produce a single point of light—a phosphene—ideally with a low threshold of activation and a brightness that corresponds to the amplitude of the electrode. If each electrode produces a single, isolated point of light, it would allow a visual image to be recreated using a pixel-based approach, assembling phosphenes into objects and images similar to an electronic scoreboard (Fig 1) [51, 52].

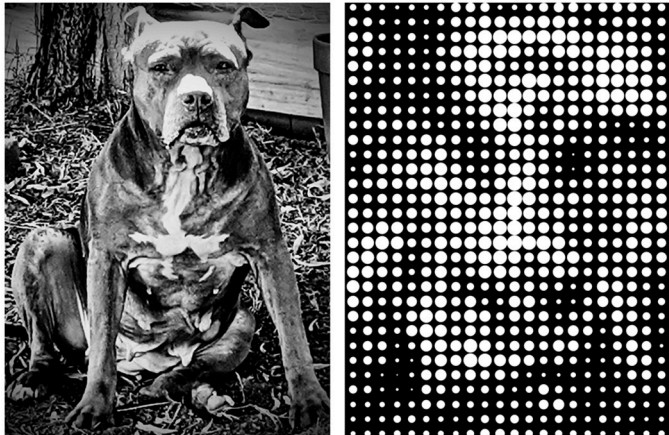

**Fig 1. Scoreboard model of pixel doctrine.** (HK).

While many early-stage research studies have reported that implant-elicited percepts are quite variable in appearance [18, 53–56], the narrative of punctate, light-colored phosphenes and the scoreboard model dominated in the literature until only recently [51, 52, 57]. Indeed while projections have been updated with more nuanced knowledge of the effects of electrical stimulation on different cell types in various stages of degeneration and reorganization [58, 59], the devices and rehabilitation protocols are still designed and built on the assumption that the quality of artificial vision produced depends on the spatial acuity of the array, as determined by electrode size, number and spacing, where each electrode will ideally produce a circumscribed phosphene.

During the course of our observations we would explicitly ask the recipient to describe the percept associated with stimulation, either during the protocol or after the session was over, and found there to be significant variability and ambiguity in this re-educative process. In UK recipients who were explicitly asked by the author (CED) during activation, many reported "glitter" or "sparkles" during single electrode stimulation. One recipient called their percepts "cheese puffs" that whizzed by laterally; another compared them to "exploding, pink popcorn;" for yet another they appeared as red diamonds, sometimes in a cluster, sometimes single (even if only one electrode was being activated). French recipients met by the author (HK) usually stuck with the terms the rehabilitation specialists used: "flashes" or "signals", or "flickering lights" ("*clignotements*").

Sometimes the phosphenes were obvious to the recipient right away, as in the case of the Recipient 1, yet still difficult to describe:

Therapist M.: [electrode activation] Do you see anything?

Recipient 1: Yes!

Company researcher S.: Please describe it

R1: Half circles within circles. Quite bright, yellow, moved to the right. . . (he indicates with a passing index finger through the air). Almost a crescent shape, with a halo around it. . ..

M: [using a list developed by the company with a list of potential descriptors] Is it is as bright as the sun, bright as a lightbulb, a candle or a firefly?

R1: . . .not as bright as the sun, but brighter than a light bulb.

[They move down the list of possible descriptors and hand him a few tactile boards that the company constructed to aid in the recipient describing size and shape. Each board has three possible size choices in one of three shape choices: 2cm, 3cm and 5 cm circles, oblong circles or rods (Fig 2).]

Recipient 1 feels his options and declares that it did not resemble any of the options exactly, but if he had to choose it most closely resembled an oblong rod and was the biggest of the

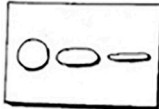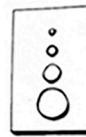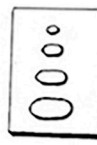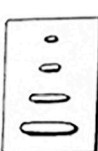

**Fig 2. Perception evaluation boards.** To assist recipients in describing their phosphenes during initial activation, they were presented boards with different forms carved out: circular shapes and sizes for the blind person to touch and choose from. (HK).

three, maybe 5cm at arm's length. They then moved down the rest of the list, giving him various options for the appearance, with Recipient 1 struggling to pick the descriptor which fit his experience best.

Then they stimulate again on the same electrode with the same stimulation parameters.

Recipient 1: Yep, I saw something (he describes it with the options they provide him. It is oblong, maybe even rectangular; again, it moves to the right, was dimmer than last time, and was 2cm—smaller than the first)

M (moving down the checklist): was it flickering?

R1: yes, both were flickering.

R1: [They activate a third time and go down the list of descriptors] half circles within circles, moving to the right, yellow, bright as a light bulb, 2cm, flickering.

Next they move onto the second step of the protocol: 10 consecutive stimulations on the same electrode. After each stimulation, Recipient 1 must answer yes or no, whether they are applying electrical stimulation or not. "Yes or no," they ask him, "do you see anything?" He gets 10 out of 10 correct, each with resounding "no" and "yes."

For others the initial activation may provide a signal that is more ambiguous or difficult for the recipient to identify.

(translated from French)

Therapist: "Here we're going to stimulate the implant a little, and you're going to tell us what you perceive." "Sometimes you won't see anything and that's normal, just let us know."

The exercise starts:

"beep" The sound indicated a stimulation has been sent to the implant.

Recipient 2: "Nothing. I hear the sound but that's all."

"beep"

R2: "I felt something in the front."

"beep, beep"

R2: "Something. . .a little flash."

"beep"

R2: "Something on the right."

"beep"

R2: "Still on the right."

"beep"

R2: "A little flash here."

"beep"

R2: "Yes."

T: "What about here?"

R2: "Mmm. It can't be on the left, right? Since the implant is on the right."

T: "Yes, it can."

"beep"

R2: "Here it's clearer." "Small flash."

"beep"

R2: "Even clearer."

(. . .)

R2: "This is harder." "The flashes are smaller."

Company researcher: "That's normal, we're starting with the lower thresholds."

(. . .)

In this situation, Recipient 2 struggles to answer the exercise with "correct answers", first by trying to locate the sensation. Experiences of this type show how unclear the sensations are, and how difficult it can be for recipients to learn to recognize what the perception is "supposed to be like". This ambiguity can be stressful for recipients, further complicating their perceptual experience with the pressure and concentration required. Within the session, the uncertainty seemed to worry Recipient 2, who repeatedly questions the therapist about the final aspect of the sensations he would be able to expect with time, demonstrating how this can affect recipients.

R2: "Is this what it's going to be like, later on?"—He asks for the second time. . .

R2: "Looking makes me tired. And when it's very small it becomes really hard!"

(. . .)

The company researcher asks him about his perception:

"Do you have a sensation?"

"Yes." Vincent answers shortly.

Then he asks yet again about what kind of perception he'll have later on. "I'll be seeing shadows, is that right?" The rehabilitation therapist tells him that he'll have to learn to "integrate" and associate the flashes. Recipient 2 says that sometimes he sees things but he's not sure that that's it [the flashes].

The reasons for ambiguity or uncertainty are multiple: 1) Description: on one hand it is difficult to describe one's visual experience. The qualia, or "what it is like" of conscious experience, is in many ways ineffable. 2) Discernment: It also may be difficult to discern: the signal is being produced within a "background" of visual distortion that characterizes blindness (that is, it is not a calm backdrop of darkness on which these phosphenes make their appearance, but instead can be a stormy sea of light and shadow, color and shape). 3) Difference: finally, it may also be that these signals are something significantly different than natural vision, and for that reason the same vocabulary that we use for natural vision just might not do.

## Learning to use the device

**Presentation of the device.**   Before camera activation, the device is presented to recipients and they are instructed on its use. The external component newly introduced at this stage consists of the visual interface, a headset made-up of opaque glasses with an integrated video camera, and a "pocket computer," or a visual processing unit that is housed in a little black box that is connected to the headset by a cable. The processing unit is about 4–5 inches and can be hung around the neck, carried in the pocket, or attached to the belt, and has various control switches that allow the device to be turned on and off switched between the different perceptual modes (i.e. depending on the device there are between 3 to 4 different image processing modes e.g. white-on-black, inverse (black-on-white), edge detection, and motion detection).

**Bodily techniques: "Seeing through the camera".**   After presentation of the device it is explained to the individual that they will be required to utilize certain bodily techniques in order to use the device—alignment of their eyes and head with the camera and scanning movements of their head. That is, the camera is effectively their new eye, and so an awareness and alignment of their head, camera, and eyes is essential to orient themselves in space via the signal. They are first taught to try and keep their eyes pointed straight with respect to their head position, using the analogy of a hand-held telescope.

Second, they are told to practice training the camera on whatever they wish to look at in space. Because the camera is not where their eye or pupil is—instead located a few inches away, in the middle of their brow ridge (above the nose, between the two eyes)—they must learn to adjust all movements and estimations of objects in space by those few inches. The trainers often tell the recipients to draw a line from the camera to the object in space with their index finger, to get the hang of the discrepancy.

Lastly, the recipient is instructed on how to move their head to scan the environment. This head scanning serves two purposes. The first large scanning movements allow a recipient to get a sense of the space and objects around them. The other is because the percept fades if the image remains stationary. Indeed, the retinal cells adapt to the stimulation pattern on the retina after a few seconds, resulting in the recipients having to make constant scanning movements with their head in order to move the camera and refresh the image. The naturally sighted viewer accomplishes this "refresh" via microsaccades—tiny movements of the eyes of which we are unconscious.

Thus, when encountering new environments, the recipients are encouraged to begin by making large scanning movements, moving their head to the farthest possible reaches in each direction—to maximize their perceptive range—followed by increasingly small movements, to refresh the image as they zoom in on an object or certain features of interest. As depicted in Fig 3, the visual field covered by the implant is quite reduced—no more than 20 degrees (about the width of two hands, outstretched), and so the recipient must scan the environment, recomposing their partial views within their minds eye. Using the device thus requires the recipient to use each bodily movement with the goal of capturing an optimal signal of the device; a process that is not intuitive to the recipient, thus requiring them to rethink the concept of "seeing."

**Camera activation.**   Two weeks after the initial activation and systems fitting, the camera is turned on. The recipient is told that this is when they will begin to gain back a kind of functional vision, and so it is a time that is often greeted with a lot of excitement. News media and camera crews who are interested in the sensational aspect of these devices are often told to come to this session.

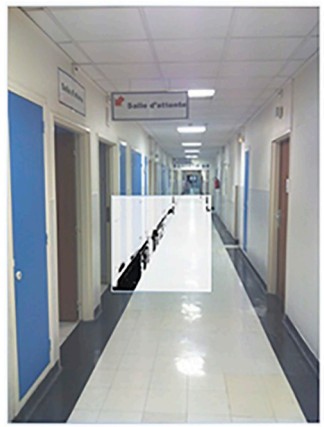 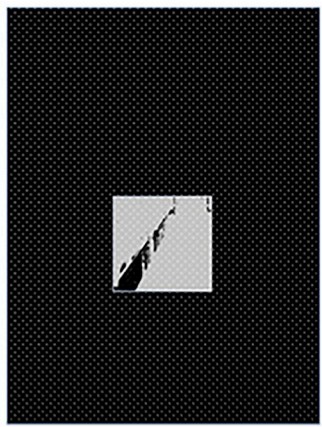

**Fig 3. Square representation of the vision field accessed through the prosthesis.** On the left you can see that it's only a portion of the whole image, and on the right you can see how the "square image" might be perceived through the device (prior to being converted into electrical pulses that would result in phosphenes) (HK).

When the camera is activated and all of the electrodes can be activated together, we found recipients reported that things became much more chaotic and "noisy." The mass of flashing lights coalesces, and the vocabulary recipients use when describing their percepts focuses on changes in the overall signal: e.g. "stronger," "more signal," "busy," "calm".

The first task that is performed when the camera is activated is tracking and localization—often with a piece of white paper, or the beam of a flashlight on the wall of a dark room. The recipient is asked to indicate if and when they see the light, and if possible, in which direction it is moving. This process is also marked by significant ambiguity.

(translated from French)

Company researcher: Could you tell us when you start to have luminous signals?

(. . .)

Therapist: Move your head a little bit downwards. Do you see something now?

Rehabilitation therapist passes the sheet in front of Recipient 3.

Recipient 3: I see luminous signals

T: Are they moving?

R3: It's blinking a little.

T: It's blinking a little?

He says while he passes the sheet in front of Recipient 3 again.

R3: Yes. Here again.

T: Ok. And when you say that here, they are doing it again, does it mean that its moving, that it's stable, that it's in front of you?

R3: Yes, it's in front of me, it appears and then it disappears.

(. . .)

Exercise is repeated another time.

T: Could you describe what you saw?

R3: Sort of a curvy shape.

T: Ok. Was there movement?

R3: No.

T: And if you had to describe it?

R3: It's rather round and its blinking.

(Therapist passes the sheet in front of Recipient 3)

R3: Here, another signal

T: It's just a flash, a light in front of you?

R3: Yes, just a flash.

(. . .)

The movement of the paper is associated with **something**–a "flash" . . .in a "curvy shape." This "something" is the first step. With the therapist's guidance, suggesting certain expressions to describe the sensations, the individual learns to define "movement" with the device associated to a sensation appearing then disappearing in different spots, and hence comes to recognize its trajectory. The main idea is that with time, the individual will learn to identify shapes. The hope is that the flash(es) that correspond(s) to the paper will be different than flash(es) associated with a different object; that over time, an individual will develop **a lexicon of flashes** corresponding to various shapes and objects. The next step is to build out this lexicon.

**Building a lexicon: Simple geometries.** The first phase of this learning protocol takes place in the radically simplified context, where the recipients are seated in front of a computer screen or a table covered in a black cloth. This simplified, high contrast situation is considered an ideal environment for the device in which they use a "building blocks" approach, inspired by simple geometries, to learn to identify simple shapes that they will later use to "decompose" more complex visual spaces. That is, this training is based on the logic that the visual environment can be deconstructed into a series of simple geometric shapes that can then be assembled into the mind of the individual and reinterpreted into a coherent visual scene.

The first exercises consist of presenting simple shapes to the recipients and have them learn to use their bodily techniques to first locate the objects, and later to identify those objects. In a typical training task, the trainer will place an object—say a white styrofoam ball—in the middle of the black table, and then instructs the individual on using the eye, head and camera alignment and head scanning techniques, giving them hints and reminders until the individual is ready to locate the object, by reaching out and touching it. Through repeated trial and error attempts, the individual learns to interpret the signals they are receiving in conjunction with the movements of their head. The recipients are also handed the ball, encouraged to get a sense of how "ballness" corresponds to the signals they receive. Over the course of a session, different-sized balls are used, progressing to different shapes (ball versus rod, ball versus ring, etc.– Fig 4), and then low vision computer monitor tasks (e.g. grating acuity–Fig 5). Through associative learning, the recipient learns to pair the kind of signal they receive with a certain shape, a skill which they can later use to decipher the environment.

Here again, for some recipients these tasks are easier or more straightforward, usually depending on whether the signals they are receiving intuitively resemble, or take the shape of their previous visual memories of the objects (i.e. if the signals they receive in association with

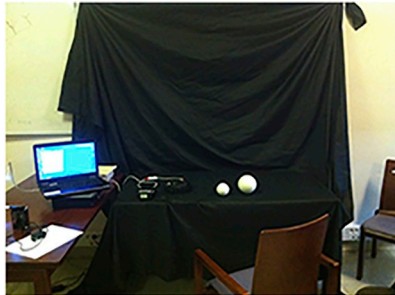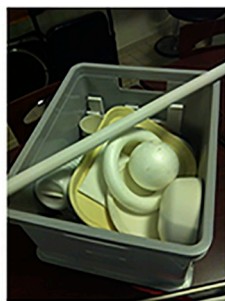

**Fig 4. Pictures of a training context for the first exercises.** Recurring shapes used during these exercises are: rectangles (or a sheet of white paper), circles (or a white ball), squares, half circles (or a banana), and later during what is called the "grating test," an acuity measure that is performed regularly throughout the protocol, as it is used as a reference point and outcome measure. (photos by HK).

the ball are "ball-like," or if lines are "linear."). For some, there is a larger discrepancy between the stimulus and their visual memory.

On whether the gratings look like lines, Recipient 4 one of the more "successful" recipients describes the way he remembers lines to look:

> "It's not what you remember. . .[instead] you learn to identify [the grating lines] with 'linear' because you know that's the way it's supposed to be."

**Decomposing space.** The recipient is then asked to put these skills of simplified geometries to work during the second phase of the training, in orientation and mobility tasks. They begin in the hallway outside of the training room, where they are encouraged to rethink the environment through an arrangement of lines. Recalling the vertical, horizontal and diagonal lines they were taught to identify on the computer screen, individuals are led to reconstruct space mentally, according to the basic angles composing it. The vision therapists are told by the companies to assume that the hallway is transformed by the device into a high contrast, black-and-white scene (Fig 6) and they coach them accordingly, encouraging them to look for the lines of the hallway and its borders, as well as of the walls interspersed with the rectangles of the doors.

During reeducation, recipients navigate down the hallway, following one of the black lines on the side. The vision therapist walks along beside the recipient, tracking the translation of the video camera image into electrode activation on a laptop they tow alongside them on a wheeled walker. The idea is to help recipients recognize key elements of the environment that they can then associate with previously learned content in order to guess the object it could likely represent. This is most often done using the previously learned line strategy. For example, if the person is in an urban environment and following the edge of the sidewalk, when the "signal" appears, the series can be reduced to the following group of possibilities: "pole",

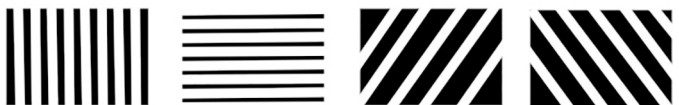

**Fig 5. Examples of screen representations for the contrast grating acuity test.** The recipients are asked which of four directions the lines are pointing, using progressively narrower spacing. (HK).

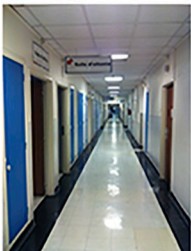 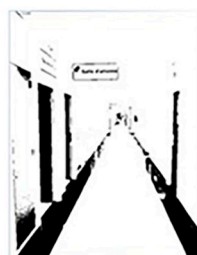 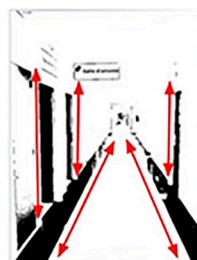

**Fig 6. Decomposing visual space according to lines.** The rehabilitation therapists are told by the companies to assume that the hallway is transformed by the device into a high contrast, black-and-white scene. The rehabilitation specialist then uses this visual to help guide the implant recipient (graphic representation by HK).

"post", "tree". If they direct the video-camera upwards, they will be able to decipher objects usually situated above, such as branches at the top of a (vertical) tree, or a sign at the top of a (vertical) post. Having a very small visual field, means that recipients must try to "follow the line" with the device—added to the continuous small head movements imitating micro saccades—which makes it all the more difficult for recipients to visualize the "line" as a whole (Fig 3).

**Recomposing the environment: Deciphering the "puzzle".** The recipients are asked to put all of the skills and techniques they have been developing together in order to "recompose" the environment. They are encouraged to practice at home, where they can rely on familiar contexts, to identify known objects around their house. It is more difficult when going outside to try and recognize new or unfamiliar objects. The recipients need to use their previously acquired blindness skills and multi-sensory abilities to get themselves in the right spot, and then use the head scanning motions and deductive techniques to detect an object in front of them (e.g. a pole). They may be able to detect the shape, and depending on contextual cues, make a guess at what it is. It is often only by pairing the new information from the implant with other senses—auditory cues—that the individual is able to make any sense of it.

For example, deciphering a car in an environment involves pairing the signals associated with "headlights" with the sound of the car approaching or driving off; for one recipient deciphering a "sidewalk" consists of pairing the feel of the sidewalk with the "shimmering lights" that were associated with the sunlight reflecting off the line of cars parked next to it.

Early on, sensory signals may contradict each other; this new sensory signal may interfere with auditory or tactile information that the recipient had adapted to use to navigate, interfering with their navigation [19]. Thus the multisensory training also takes time, as the recipient learns to suppress or realign certain senses with the new signals they are receiving. Recipients describe the recomposition, or reinterpretation, of the signals as the most difficult part of device use. While the signals can be considered "visual" as far as they consist of light and sensation at a distance, because they are so different from what the recipient remembers of visual appearance, it can take considerable time and effort to interpret. In certain cases, the signal remains so ambiguous—or can interfere with integration of the other senses—that the recipient will never be able to use it in a complex visual scene, certainly not with the device by itself.

"Seeing" is thus a process of recomposing and deciphering the tangle of signals they get from the device, integrating with other sensory signals and visual memory. How do recipients describe seeing an object? Recipients might explain "I see a car," or "I see a tree"–as is publicized in the company-based videos and publications—but if asked to describe their process, one begins to get a sense how multimodal it is.

Two recipients describe this (translated from French):

"What surprised me most were the cars. (. . .) There's plastic in the front, the hood is metal, and then the window and. . .. they are not equally luminous. (. . .) The bumper is in front and the hood is horizontal, and the window is like this [she says drawing a diagonal with her hand]. So it doesn't reflect light in the same way. (. . .) When one sees a car [with natural vision] it's a whole, a shape. And you can see it as a whole. But **I** have to put it back together like a puzzle. That, plus that, plus that (. . .): makes a car. It's weird at first, but now I'm integrated in this way."

—Recipient 5

"For a tree, I don't see the trunk, only a vision of the tree, so I see something like a vertical line. I move my head a little, and see a vertical line. . .then, while zooming like this [with the controls], I move my head upwards, and then I see full of little flashes. So then I know they are branches."

[he explains that flashes = leaves]—Recipient 6

## Living with the device

**What artificial vision is "like".** One can think about this in terms of the "gestalt" of artificial vision. The recipient is faced with a mass of flickering flashes on a background of shifting light, shadow and shape (depending on their Charles Bonnet Syndrome, condition in which individuals who have lost their sight experience visual hallucinations). Through this periscope view of the environment, and through a series of movements of the head that they use in conjunction with their visual memory and other senses, recipients try and construct a cohesive picture of the visual environment that they can then interpret. All visual stimuli become reduced to flashes of varying intensity occurring in two-dimensional space (i.e. there is no depth information). The act of attending to this shifting visual scape (and pairing it with reassembling and interpretation) is something that requires intense focus and concentration, requiring that the individual direct their attention simultaneously to a large variety of sensory information from different senses at the same time.

We purposefully did not identify and differentiate between the two different devices here because we did not find recipient experience to vary significantly by device. That is, while we did find there to be significant inter-individual variability in perceptual experience, whether it was Argus II or IRIS II did not seem to matter, and this was despite the difference in number of electrodes (60 vs 150, respectively). This is in line with published literature on clinical outcome measures comparing other visual prosthesis devices—the Argus II and the Alpha IMS, which not only differed in terms of the number of electrodes (60 vs 1500) but also in terms of location on the retina (epiretinal vs subretinal), without any significant difference in clinical outcomes between the two (despite a projected 7-fold increase in acuity by the Alpha IMS) [23].

**Artificial vision is electric.** Overall, across all recipients, artificial vision was described using terms that invoked electric stimuli. English recipients most commonly reported a "light show," "lots of flashing lights" and/or "fireworks." In French recipients the most recurring words are "*clignotements*" (translates into "flashing", or "blinking lights"), "*signaux lumineux*" ("luminous signals"), "*petits flashs*" ("small flashes"), "*scintillement*" ("shimmering"). Recipients frequently used metaphors to try to describe these perceptions: "Eiffel tower lights"; "Christmas lights", and "camera flash."

**Artificial vision is "different".**   When asked how artificial vision compared to what they remember natural vision to be like, many recipients responded with a variation of what one recipient, Recipient 7, reported: "first off, you need to understand that it is fundamentally different than natural vision." Indeed, one vision rehabilitation specialist who had worked with scores of Argus II recipients said that the overwhelming response is that artificial vision is "different" than the individual expected to be. She stated that she has yet to meet someone who reports the experience was exactly what they expected it would be, even after their expectations are tempered in the process of being vetted for the device.

**Use (or disuse) over time.**   Over months to years, outside of the rehab facilities and clinical trial testing rooms, recipients reported a variety of experiences. The device can fail prematurely (as in case of IRIS I & IRIS II) or keep working for the expected lifetime of the device (durability studies in Argus II show over 8,5 years) [60]. Some recipients report that after they use it to get a sense of the familiar objects in their own home, they feel they do not have a use for it, and report having a "so what" stage after 1–2 years, where individuals are disappointed by the device. For this reason, many recipients just stop using their device after a spell. One researcher said of the 12 recipients he had worked with on a study, two years later none of them used the device. These observations join those on experiences with other prosthetic technologies such as cochlear implants or prosthetic arms, often abandoned (temporarily or definitively) because of their inconvenience and unnatural qualities [33, 61].

Indeed, as with prosthetics mentioned above, of everyone we spoke to—even the companies 'banner' recipients—all mentioned that the process of using the device in daily life never ceased to be intensely cognitively fatiguing: both because of the continuous and intense focus they must invest in what becomes an actual perceptive activity (natural vision usually does not require such effort) and because of the nature of the sensations produced, inherently different from "natural vision" as we know it. According to one rehabilitation specialist who has worked with dozens of device recipients, it is likely for this reason that a majority of recipients eventually discontinue use of the device not long after their training period is done. Recipients who do continue to use the device tend to stick to the "contemplative function" aspect of the device —using it in conditions that are not "too bright," nor "too busy"- and for tasks that are not of consequence. That is, some recipients spoke of a certain "pleasure" of "watching" the leaves of a tree shimmer, or perceiving how high the Louvre pyramid is (it was not built until after that particular recipient lost their vision). Recipient 8 reported that he only used the device for skiing, an activity that he was passionate about; not so much to help him ski (as he relies on his skiing partner and not on the device), but rather to have additional sensation during the experience.

**Percept does not change over time.**   All recipients, and all vision rehabilitation specialists who worked with the recipients, were clear about the fact that the quality of the percept does not change over time as the individual learns to use the device—and this extends over the course of years for some recipients. That is, regarding one of the most important questions for the development of these devices—whether the individual can transform the imperfect signal through perceptual learning over time—we learn that no, they do not seem to. Instead, one just learns to interpret the signal that is provided.

**Psychosocial effects.**   Something that all of the recipients we spoke to brought up in terms of their experience—often without prompting—was the significant psychosocial effects they encountered by participating in the trials. These effects can be considered as threefold:

First, recipients can be disappointed in the beginning by the difference of artificial vision from what they were expecting. Even with a change in the rhetoric the researchers learned to employ over time, stressing that the kind of vision these recipients would get would be

different than natural vision they remembered, the recipients all said they were unprepared for just how different it was.

Second, a common theme that arose time and again was the way in which recipients treat failure of the device as a personal defeat. Whereas recipient successes are claimed by the company to be a product of the device, failures are more often than not put upon the shoulders of the recipients. For example, we have the case of Recipient 9. When Recipient 9's device stopped functioning he was told that it was "his eye" that was not working and not the device, a projection of responsibility that he took personally and found to be unfair, as no evidence was presented to him of why his eye should stop working. Or in other cases where, when progress stalls, the recipients are told it is because they need to be practicing more at home—that it is their brain that sees and constant training is needed for this to happen.

Finally, there are also the psychosocial effects linked to both benefits and difficulties associated with the change in social relationships experienced in getting one of these devices. It seems that one of the biggest benefits of getting one of these devices that recipients and researchers alike spoke of—whether the device works or not—is the job, role and purpose it gives recipients: to receive attention, to have a use, to be surrounded by a community. This ethos was reflected by a clinical coordinator at one of the companies: "we're giving these individuals a job, a purpose, and they all really responded to that. . . these individuals are given attention and a community of supporters that revolve around them; you have a research group who is indebted and grateful to you for your services. That is what you get out of participating in the trials." But then, when the device stops working or the trial concludes—one loses all of this; not only one's new sensory relation with the world, but also the role, the job, the identity, the community. More than one recipient talked about the difficulties encountered when the device stops working. "It is like going blind and losing the possibility of sight all over again" one recipient, Recipient 4 reported.

It is notable, however, that even in recipients who were disappointed by the quality of the perceptual experience they received, and in spite of the psychological difficulties—everyone we spoke with wanted to be considered for the next generation device. In many cases they were hesitant to discuss their difficulties or complaints because of their belief that it might endanger their consideration by the companies for the next generation device.

## Discussion

We undertook ethnographic research with a population of retinal prosthesis implant recipients and vision rehab specialists, documenting the process of getting, learning to use and living with these devices.

We found that the perceptual experience produced by these devices is described by recipients as fundamentally, qualitatively different than natural vision. It is a phenomenon they describe using terms that invoke electric stimuli, and one that is ambiguous and variable across and sometimes within recipients. Artificial vision for these recipients is a highly specific learned skillset that combines particular bodily techniques, associative learning and deductive reasoning to build a "lexicon of flashes"—a distinct perceptual vocabulary—that they then use to decompose, recompose and interpret their surroundings. The percept does not transform over time; rather, the recipient can better learn to interpret the signals they receive. This process never ceases to be cognitively fatiguing and does not come without risk nor cost to the recipient. In exchange recipients can receive hope and purpose through participation, as well as a new kind of sensory signal that may not afford practical or functional use in daily life, but for some provides a kind of "contemplative perception" that recipients tailor to individualized activities. We expand on these findings below to explore what they mean in terms of the

development and implementation of these devices, as well as for our understanding of artificial vision as a phenomenon.

What does it mean that the recipients describe artificial vision as being fundamentally, qualitatively "different" than natural vision? We believe that acknowledging that artificial vision is a unique sensory phenomenon might not only be more accurate, but it may also open up new avenues of use for these devices. Artificial vision may be considered as "visual" in terms of being similar to what recipients remember of the experience of certain kinds of light, as well as by offering the possibility of being able to understand features of the environmental surround at a distance. That being said, artificial vision was also described as both qualitatively and functionally different than the "natural" vision the recipients remember. It is in this way that the sensory experience provided by these devices could be viewed as less a restoration or replacement and more as a substitution; that is, as offering an entirely different or novel sensory tool. By shifting from the rhetoric of replacement or restoration to substitution we believe it could widen the bounds in which researchers and rehabilitation specialists think and operate with regard to how these devices are designed and implemented, potentially liberating a whole new spectrum of utility through the novel sensations these devices produce. Likewise, this shift could change the expectations of individuals receiving these devices, including addressing the initial disappointment that was expressed by many of our recipients when they encountered just how different the signals were to what they were expecting.

Second, acknowledging artificial vision as a unique sensory phenomenon also helps us understand the importance of qualitative description. The process of learning to use the device is a cooperative process between the rehabilitation specialist and the recipient, with the specialist guiding the recipient to attend to their perceptual experience and interpret it in specific ways. This process begins with the recipient learning to recognize how the basic unit of artificial vision—the phosphene—appears for them, and then describe that to the rehab specialist. The specialist then uses this information to guide the recipient in learning how the phosphenes correspond to features of the environment. It is a continuous and iterative communicative practice between the recipient and specialist that evolves over many months, during which stimuli are encountered, the recipient responds, and the specialist gives corrective or affirming feedback (with more or less description by the recipient and guidance by the specialist depending on the dynamic and need). The process is so specific to the dynamic between recipient and specialist that it can be considered to be "co-constructed" within their interactions.

Because each recipients' qualitative experience is so distinct (phosphenes differ significantly between recipients so that no recipients' perceptual experience is alike [60]) each process is tailored to the individual recipient by specific specialists. We found that certain vision rehabilitation specialists inquire in more depth about a recipient's qualitative experience than others, using different methods, styles and techniques, and this can result in a different experience—and thus outcome—for the recipients. Our findings are based on reports captured either by directly asking the recipients about their experience or observing descriptions that were part of the rehabilitation process but that were by and large not recorded by the specialists nor relayed back to the companies, early stage researchers nor the individuals being implanted. That is, we found that there is no protocol in place for capturing or sharing recipient's qualitative reports, including within the companies (between various clinical sites). Yet these kinds of data are essential to understanding these devices as well as in learning about artificial vision more generally, and thus deserve careful consideration by both researchers and clinicians who are developing and implanting these and similar devices, as well as to individuals and their families who are considering receiving them.

The better vision rehabilitation specialists are able to understand the recipient's qualitative experience, the more they are enabled to assist them in learning to use the device. The more

that early-stage researchers understand about how the parameters of the device correspond to perceptual experience, the better they are able to optimize design and implementation strategies. Finally, communicating these data to individuals and their families who are considering being implanted is essential. It would contribute to a more accurate understanding of the qualitative experience and process they are signing on for, and thus is an important part of informed consent. It would also help to address certain psychosocial difficulties we found recipients to experience. For instance, we found that the recipient's percept does not change over time—that instead the recipient becomes better able at interpreting the signal received using cognitive techniques. It is a subtle distinction, but a profound one in terms of conditioning expectations around these devices—both of the researchers and the recipients. We found that current rhetoric employed by researchers and vision rehab specialists regarding neuroplasticity and the ability for recipients to transform the signal with enough practice has created a situation in which failure of the recipient to significantly transform the signal over time is perceived as a failure of the recipient (behaviorally, where the recipient is deemed to have insufficiently practiced using of the device, and/or physiologically, where the problem is located within the recipient's eye or visual system). By shifting the expectation that it is not the percept itself, but the recipient's ability to use the percept over time that can improve, one can potentially avoid and address the psychosocial distress that we found some recipients experienced as a result.

This study had several limitations, first and foremost the number of recipients limited by small study populations and availability of recipients. Future studies of these devices would do well to include similar qualitative reports from recipients, either as primary focus or as supplement to other outcome measures (i.e. as "mixed methods" studies that combine qualitative and quantitative methodologies). In addition, qualitative reports are only one type of data and are not meant to replace other forms of data being collected on these devices. Rather, we believe they deserve special attention because they have been heretofore neglected in the literature despite their potential to provide valuable information not captured by normative functional outcome measures. Qualitative data about recipients' perceptual experience can both inform device design and rehabilitative techniques, as well as grant a more holistic understanding of the phenomenon of artificial vision. In addition to contributing to the larger body of work on visual prostheses, this study serves as a case example of the kind of data mobilized by qualitative, ethnographic methodology—in particular phenomenological inquiry—in study of brain machine interface devices.

## Acknowledgments

The authors would like to individually thank all the anonymized persons from the hospitals' personnel, as well as the people interviewed working for companies Pixium and Second Sight, and finally and most importantly the retinal implant recipients who graciously accepted to be observed and interviewed.

## Author Contributions

**Conceptualization:** Cordelia Erickson-Davis, Helma Korzybska.

**Data curation:** Cordelia Erickson-Davis, Helma Korzybska.

**Formal analysis:** Cordelia Erickson-Davis, Helma Korzybska.

**Funding acquisition:** Cordelia Erickson-Davis, Helma Korzybska.

**Investigation:** Cordelia Erickson-Davis, Helma Korzybska.

**Methodology:** Cordelia Erickson-Davis, Helma Korzybska.

**Project administration:** Cordelia Erickson-Davis, Helma Korzybska.

**Resources:** Cordelia Erickson-Davis, Helma Korzybska.

**Software:** Cordelia Erickson-Davis, Helma Korzybska.

**Supervision:** Cordelia Erickson-Davis, Helma Korzybska.

**Validation:** Cordelia Erickson-Davis, Helma Korzybska.

**Visualization:** Cordelia Erickson-Davis, Helma Korzybska.

**Writing – original draft:** Cordelia Erickson-Davis, Helma Korzybska.

**Writing – review & editing:** Cordelia Erickson-Davis, Helma Korzybska.

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
