## [Decision Letter · Decision Letter 0]

24 Jun 2020

PONE-D-20-02730

What do blind people “see” with retinal prostheses? Observations and qualitative reports of epiretinal implant users

PLOS ONE

Dear Dr. Erickson-Davis,

Thank you for submitting your manuscript to PLOS ONE. After careful consideration, we feel that it has merit but does not fully meet PLOS ONE’s publication criteria as it currently stands. Therefore, we invite you to submit a revised version of the manuscript that addresses the points raised during the review process.

I want to profoundly apologise for the delay in obtaining reviews for you. That it was harder to obtain reviewers for your paper is a reflection of current circumstances, rather than the interest in your paper. As I hope you agree, their suggestions are thoughtful and should improve the value of your paper to the field. Both reviewers noted the need for more details about the participants, how they were interviewed, and how representative of the cohort their answers might be. I do not believe this requires detailed quantitative analysis; rather further explanation in the text should be sufficient to address the reviewers' concerns. 

We look forward to receiving your revised manuscript.

Kind regards,

Nicholas Seow Chiang Price, Ph.D.

Academic Editor

PLOS ONE

Journal Requirements:

Please explain why was written consent was not obtained, how you recorded/documented participant consent, and if the ethics committees/IRBs approved this consent procedure.

3. Please include a copy of the interview guide used in the study, in both the original language and English, as Supporting Information, or include a citation if it has been published previously.

Reviewers' comments:

Reviewer's Responses to Questions

**Comments to the Author**

1. Is the manuscript technically sound, and do the data support the conclusions?

Reviewer #1: Partly

Reviewer #2: Partly

2. Has the statistical analysis been performed appropriately and rigorously? 

Reviewer #1: N/A

Reviewer #2: I Don't Know

3. Have the authors made all data underlying the findings in their manuscript fully available?

Reviewer #1: No

Reviewer #2: Yes

4. Is the manuscript presented in an intelligible fashion and written in standard English?

Reviewer #1: Yes

Reviewer #2: Yes

5. Review Comments to the Author

Reviewer #1: 1. It is difficult to get a sense of how technically sound the manuscript is, and how well the data support the conclusions drawn, because there is some vagueness surrounding transparency of the data, and how the data were analysed. It is not possible, for instance, to know whether specific sentiments reported in the manuscript were drawn from a single interview, or aggregated from multiple sources. Consider ways in which the manuscript might be able to indicate how gathered sentiments regarding various aspects of artificial vision are shared across the population included in the study.

2. No statistical analysis was performed. Not applicable.

3. No. While the authors appear to claim that they have made all data publicly available, patient data remains confidential. If possible, the authors should de-identify data and make it publicly available. This would also support question 1.

4. Yes. The manuscript is excellently written and easily intelligible.

Reviewer #2: This is an important and needed paper. I worked with retinal prosthesis persons and the patients’ reports were often disheartening given their disappointment and the consequent implant disuse. I commend the authors for finally writing a qualitative report of what these patients feel and how/whether they find the new information useful at all. However, the manuscript could be improved in many respects and thus I have suggested a number of minor and major revisions below:

Line 85: “Meanwhile, there many other” I believe should be ‘there are’

Line 90: “impacted by RP” please define RP the first time you use it

Throughout the manuscript do not use abbreviations, e.g. isn’t should be replaced by is not, don’t by do not etc…

Line 126-127: “as well as to our understanding of vision more generally.” How does adding subjective experience help with our understanding of vision more generally? This statement needs a bit more elaboration as the vision afforded by these prostheses is quite different from typical vision so what knowledge do this study add to our understanding of vision more generally? This is unclear in the introduction but also in the discussion

Line 158 to 167: “I believe this type of information about the authors is unnecessary here but can be moved to the supplemental material

Line 212-214: “Nine of these participants will be presented in this article, we anonymously called them: Douglas, Vincent, Isabelle, Arthur, Eve, Thomas, Danny, Benoit, and Mathew.” Not sure I follow this, if participants were 16 why 9 of these participants are presented? Or are these additional to the 16? Please clarify, also why using names at all you could just use participant 1, 2, 3 etc…as better way of maintaining participants anonymous

The information about the Interview is quite vague and would not allow replication. Please add all the interview used in the supplemental material and refer to it in the main manuscript, also report few example questions also in the main manuscript to explain what type of interview was/what type of questions were used. The same applies to the Data analysis, please give more information, e.g. “These data were supplemented with observational data collected by the authors, translated into field notes that were then subject to qualitative coding methods [46].” Line 235-236, how were the data supplemented by observational data exactly and what qualitative coding methods? The method and analysis should be clear and detailed enough to allow for replication

It would be good to add a figure with a photo of the two implant systems examined in the present paper somewhere relevant in the manuscript

Line 250-251: “To be eligible to receive one of these devices the individual 250 must have bare light perception or no light perception (in order to warrant the risk of the device further damaging any residual vision).” Is this the only eligibility criterion?

Line 260: “techniques)” full stop is missing

Line 274: “in particular the psychological profile individual subjects” I believe should be of individual subjects

Line 295-296: “which receives power wirelessly coil and electronics case either a 60 or 150-electrode array that is fixed to the inner surface of the retina.” This sentence is quite difficult to follow and understand please rephrase it

Although I agree that a qualitative study on the subjective experience of the implants was long overdue, using one method as the authors mention themselves is always limited as the link between subjective and more objective measures remains unclear, so why not using a mixed-method approach? With both qualitative and quantitative measures obtained and examined together? A discussion of this limitation and suggestion for future direction should be added in the Discussion (the authors mention this limitation but no clear suggestions for future research are given)

Fig. 2 would be good to have an actual photo of the carved stimuli used rather than a sketch

Line 384 to 386: “Douglas must answer which of the 10 they are applying electrical stimulation and which ones they aren’t. Yes or no, they ask him, do you see anything? He gets 10 out of 10 correct, each with resounding “no” and “yes.” The task is unclear here, were participants asked to answer after each stimulation if a stimulation occurred or not? Please clarify

Line 442: “one’s visual experience the way it is for anyone to describe” difficult to understand needs rephrasing

Line 454: “When individuals are ready to commence with camera activation” how is this decided and how it varies?

Line 460-462: “(i.e. depending on the device there are between 3 to 4 different image processing modes e.g. white-on-black, inverse (black-on-white), edge detection, and motion detection).” Can a figure representing all these different modes be added?

Line 482: “resulting in the participants must make constant scanning” resulting in participants having to make

Line 488: “they zone in on an object” I believe you mean zoom in

Line 493: “in this way it requires that they rethink the concept of “seeing.”” It requires them rethinking the concept of “seeing”.

Line 525: “He says while he passes the again.” He passes the sheet in front Isabelle again.

It is very useful to have a clear description of all the phases involved in the process required to deliver and use a retinal implant, however the data and subjective reports are scarce given the 16 participants tested, it would be good to have more examples of participants reports throughout the manuscript and a table with examples on a specific theme from all participants to better understand the subjective variability and experience with the implants, also it would be useful to report, the age of the participant what implant the participant was using as two are examined here when reporting quotations [e.g. quote, Participant 1, female, 56 years, ARGUS II] at the moment the manuscript reads more as a description of the retina implant process than of the subjective experience of the participants which was the main aim of the study.

Line 614-615: please use participants rather than ‘them’ twice in a sentence, e.g “The vision therapist walks along beside them, tracking the translation of the video camera image into electrode activation on a laptop they tow alongside them on a wheeled walker.” The vision therapist walks along beside the participant, ….them on a wheeled walker.

Line 694: “This accords with published literature” better “this is in agreement or in line with”, also some more references are needed in these sentence, what published literature? Is just one study? [23]? Explain how is in agreement with this study, what did they show?

Line 794: “lest it endanger” because it could endanger

Line 823-827: “While “visual” in terms of being similar to what participants remember of the experience of certain kinds of light, as well as by offering the possibility of being able to understand features of the environmental surround at a distance, artificial vision was described as both qualitatively and functionally different than the “natural” vision the participants remember.” Very long and difficult to follow sentence consider splitting it into two sentences and rephrase it

Line 829: “and more a substitution; that is, as offering an entirely different or novel sensory tool” I agree but then there are lots of sensory substitution devices (e.g. The vOICe) that are less invasive and less difficult to use and are free, so why going through all this process? A discussion of this point would be important as these implants are as you mention cognitively heavy and at least similar (if not at times better) goals can be achieved with sensory substitution devices relying on available and intact senses such as sound and touch rather than lost vision

Line 869: “The more early” the earlier

Line 876: “we found that that the participant’s” delete one “that”

In the discussion it would be good to underline how the use of these implants is discontinued by the far majority of patients and whether anything emerged in the interviews and data that could be done to improve this outcome

6. PLOS authors have the option to publish the peer review history of their article (what does this mean?). If published, this will include your full peer review and any attached files.

Reviewer #1: No

Reviewer #2: No

---

## [Author Response · Author response to Decision Letter 0]

29 Oct 2020

We thank the reviewers and the journal for their substantive read and comments.

Journal Requirements:

Response: We acknowledge that we have updated the manuscript in line with these style requirements to the best of our ability.

Please explain why was written consent was not obtained, how you recorded/documented participant consent, and if the ethics committees/IRBs approved this consent procedure.

Response: Manuscript has been revised - see lines 214-220.

3. Please include a copy of the interview guide used in the study, in both the original language and English, as Supporting Information, or include a citation if it has been published previously.

 Response: Interview guide now included as supplementary information

Response: Cover letter has been updated to indicate the following:

Please note that our datasets cannot be made publicly available as they contain both identifying and sensitive participant information. The small number of participants implanted with these devices and the nature of the interview content (e.g. the focus on individual participant life histories and candid responses) make anonymization of the data unfeasible. Participants gave their consent with the understanding that their responses would remain confidential. In particular, it was the wish of the many of our participants that the companies manufacturing the devices and running the trials not have access to their candid statements (and it would be easy for researchers working for these companies to identify the participants based on any number of narrative details included in our field notes). Confidentiality of their data is what was specified and agreed upon by the local regulatory bodies overseeing this study. We will consider specific requests for data, with special consideration to the identify of the requester. These requests may be sent to either of the authors, as well as our respective ethics committees at the addresses provided.

5. PLOS requires an ORCID iD for the corresponding author in Editorial Manager on papers submitted after December 6th, 2016. Please ensure that you have an ORCID iD and that it is validated in Editorial Manager. To do this, go to ‘Update my Information’ (in the upper left-hand corner of the main menu), and click on the Fetch/Validate link next to the ORCID field. This will take you to the ORCID site and allow you to create a new iD or authenticate a pre-existing iD in Editorial Manager. Please see the following video for instructions on linking an ORCID iD to your Editorial Manager account: https://www.youtube.com/watch?v=_xcclfuvtxQ. 

Response: ORCID IDs for both authors have been confirmed

Reviewers' comments:

Reviewer #1: 

· The paper relies on a form of analysis called Interpretative Phenomenological Analysis. It seems unclear how this analysis derives conclusions from the data. Engineers looking to interpret this analysis in order to make iterative improvements will need to understand the broad strokes of how it operates. A briefly expanded discussion of the data analysis process could greatly assist in the interpretation of results – most especially so for those readers outside the phenomenological ethnography field. 

Response: Manuscript revised - see lines 149-157 in the introduction, and lines 260-273 in the method/ data analysis section.

· While it is true that few data are available on the qualitative perception of artificial vision to date, it is not true that there is none. The paper might make brief reference to existing work and note continuity or disparity between experiences. One that jumps to mind is Jens Naumann’s book on Dobelle’s cortical implants. (https://www.amazon.com/Search-Paradise-Patients-Artificial-Experiment/dp/1479709204. 

Response: Correction and citation has been added to see lines 121-122.

· The paper makes no mention of the quantitative results of various psychophysics testing the patients underwent. It seems a missed opportunity to tie qualitative description to quantitative result – if possible, the paper should make every effort to include quantitative results as well. 

Response: The results from the psychophysics testing the patients underwent are considered proprietary data that belong to the companies. We were not granted permission to publish on these data. 

· In general, I would have liked to see a section on recommendations for future design. The interpretation of these results as is left largely to the engineers who will not understand the analysis. Greater clarity in the underlying data, how analysis was conducted, and how results should be interpreted would significantly improve this paper. 

Response: Suggestions for how these findings may both be interpreted by those utilizing these devices as well as possible future research directions are discussed in the discussion (Lines 884-951). Specific recommendations for design of the physical device itself is outside the scope of this manuscript, and the abstract and introduction have been revised to ward off that expectation (Lines 59-60). Clarification regarding analysis per above.

· The paper claims sixteen people were followed, nine of which appear in the paper. A brief note as to why seven people were rejected might improve clarity in the data collection and analysis. 

Response: All sixteen participants’ data are reflected in our major findings. There are 9 participants from whom we quote directly; the other seven were not rejected or discarded - we simply did not quote from them directly. This is the standard for ethnographic research.

· The authors contradict themselves on data availability, first claiming that all data is fully available without restriction, then claiming patient confidentiality. While it is appropriate for the authors to restrict data availability on this basis, the authors might de-identify data where possible, thereby allowing some data to be shared. 

Response: We have clarified that our datasets cannot be made publicly available as they contain both identifying and sensitive participant information. The small number of participants implanted with these devices and the nature of the interview content (e.g. the focus on individual participant life histories and candid responses) make anonymization of the data unfeasible. We will consider specific requests for data. These requests may be sent to either of the authors, as well as our respective ethics committees at the addresses provided.

· The figures presented could do more to illustrate the main findings of the paper. Perhaps cartoon drawings of cognitive processes, such as building a scene from a lexicon of flashes, might be more helpful. 

Response: We wish to refrain from putting visuals to speculation out of concern that such heuristics may be interpreted literally or be otherwise misleading. One of our key findings is that contrary to expectations that these devices would create a grid of phosphenes that could used to build a scene (expectations encouraged by illustrations in publications and pitch decks), the phosphenated scenes are chaotic and messy, and the cognitive processes difficult and taxing. 

· While well-written and presented, there remain a few typos throughout. Lines 571 and 893 are two examples that caught my eye. 

Response: Noted and corrected.

· It is difficult to get a sense of how technically sound the manuscript is, and how well the data support the conclusions drawn, because there is some vagueness surrounding transparency of the data, and how the data were analysed. It is not possible, for instance, to know whether specific sentiments reported in the manuscript were drawn from a single interview, or aggregated from multiple sources. Consider ways in which the manuscript might be able to indicate how gathered sentiments regarding various aspects of artificial vision are shared across the population included in the study. 

Response: The findings of the paper were shared across the population included in the study. The manuscript has been revised to clarify this (see lines 230-232)

Reviewer #2: This is an important and needed paper. I worked with retinal prosthesis persons and the patients’ reports were often disheartening given their disappointment and the consequent implant disuse. I commend the authors for finally writing a qualitative report of what these patients feel and how/whether they find the new information useful at all. However, the manuscript could be improved in many respects and thus I have suggested a number of minor and major revisions below:

Line 85: “Meanwhile, there many other” I believe should be ‘there are’. 

Response: Manuscript revised see line 85

Line 90: “impacted by RP” please define RP the first time you use it. 

Response: Manuscript revised see line 90

Throughout the manuscript do not use abbreviations, e.g. isn’t should be replaced by is not, don’t by do not etc… 

Response: Manuscript revised, with exception of direct participant statements

*Line 126-127: “as well as to our understanding of vision more generally.” How does adding subjective experience help with our understanding of vision more generally? This statement needs a bit more elaboration as the vision afforded by these prostheses is quite different from typical vision so what knowledge do this study add to our understanding of vision more generally? This is unclear in the introduction but also in the discussion

Response: These devices were built using the dominant theories of visual perception -e.g. Marr and Poggio’s computational theory of vision, with the expectation (voiced by many in the field - e.g. Chichilnisky, Palanker, Zrenner among others) that, with enough training, an individual would be able reconstruct an image from the phosphenes produced by electrical stimulation. As any examination of the clinical outcomes of these devices makes clear: these expectations have not been met. That the vision afforded by these devices are quite different from typical vision is informative itself. Indeed, we believe the discrepancy between expectations and experience of these participants can be used to support alternative theories of vision, which author CED discusses in a manuscript currently under review. 

Line 158 to 167: “I believe this type of information about the authors is unnecessary here but can be moved to the supplemental material. 

Response: Manuscript revised see line 172-177

Line 212-214: “Nine of these participants will be presented in this article, we anonymously called them: Douglas, Vincent, Isabelle, Arthur, Eve, Thomas, Danny, Benoit, and Mathew.” Not sure I follow this, if participants were 16 why 9 of these participants are presented? Or are these additional to the 16? Please clarify, also why using names at all you could just use participant 1, 2, 3 etc…as better way of maintaining participants anonymous. 

Response: All sixteen participants’ data are reflected in our major findings. There are 9 participants from whom we quote directly; the other seven were not rejected or discarded - we simply did not quote from them directly. This is the standard for ethnographic research. Regarding nomenclature, manuscript has been revised per recommendation

The information about the Interview is quite vague and would not allow replication. Please add all the interview used in the supplemental material and refer to it in the main manuscript, also report few example questions also in the main manuscript to explain what type of interview was/what type of questions were used. The same applies to the Data analysis, please give more information, e.g. “These data were supplemented with observational data collected by the authors, translated into field notes that were then subject to qualitative coding methods [46].” Line 235-236, how were the data supplemented by observational data exactly and what qualitative coding methods? The method and analysis should be clear and detailed enough to allow for replication

Response: Interview guide has now been added as supplementary material and data analysis section briefly expanded. We also believe this comment and question relates to the unfamiliarity the reviewer has with ethnographic methods. This confusion may shared by other readers and so the point is well taken! 

To clarify: ethnography is a kind of qualitative methodology that allows one to provide a detailed or “thick description” of a phenomena and its participants, based on many hours of direct observation and interviews with key informants. Observations are translated into field notes and interview data are transcribed. Analysis of these ethnographic data is undertaken in an inductive, thematic manner: data are examined to identify and to categorize themes and key issues that “emerge” from the data. Using this inductive process, ethnographers generate tentative theoretical explanations. It is quite different than other methods that used standardized survey technology that utilize pre-determined question’s that are asked to all participants in the same way. Ethnographers use informal or conversational interviews which allow them to discuss, and probe emerging issues or ask questions in a naturalistic manner. Because of the “casual” nature of this type of interview technique it can be useful in eliciting highly candid accounts from individuals. 

We have added additional material to the introduction clarifying what ethnographic methods are (lines 149-157) to better situate the goal and methods of this project.

It would be good to add a figure with a photo of the two implant systems examined in the present paper somewhere relevant in the manuscript. 

Response: Manuscript revised – see figures 1&2 

Line 250-251: “To be eligible to receive one of these devices the individual 250 must have bare light perception or no light perception (in order to warrant the risk of the device further damaging any residual vision).” Is this the only eligibility criterion? 

Response: Indeed not; the manuscript has been revised to remove the topic of eligibility criteria altogether as it is not essential to the points we are making

Line 260: “techniques)” full stop is missing. 

Response: Manuscript revised.

Line 274: “in particular the psychological profile individual subjects” I believe should be of individual subjects. 

Response: Manuscript revised.

Line 295-296: “which receives power wirelessly coil and electronics case either a 60 or 150-electrode array that is fixed to the inner surface of the retina.” This sentence is quite difficult to follow and understand please rephrase it. 

Response: Manuscript revised – now line 336-338

Although I agree that a qualitative study on the subjective experience of the implants was long overdue, using one method as the authors mention themselves is always limited as the link between subjective and more objective measures remains unclear, so why not using a mixed-method approach? With both qualitative and quantitative measures obtained and examined together? A discussion of this limitation and suggestion for future direction should be added in the Discussion (the authors mention this limitation but no clear suggestions for future research are given). 

Response: Manuscript revised to mention the utility of mixed-methods approach as future area of research (see lines 948-957). However, as we say, the focus of this manuscript is to focus on qualitative reports given they have heretofore been neglected. 

Fig. 2 would be good to have an actual photo of the carved stimuli used rather than a sketch 

Response: Unfortunately this was considered proprietary information and we were not allowed to take photos.

Line 384 to 386: “Douglas must answer which of the 10 they are applying electrical stimulation and which ones they aren’t. Yes or no, they ask him, do you see anything? He gets 10 out of 10 correct, each with resounding “no” and “yes.” The task is unclear here, were participants asked to answer after each stimulation if a stimulation occurred or not? Please clarify. 

Response: Manuscript revised – see lines 427-429

Line 442: “one’s visual experience the way it is for anyone to describe” difficult to understand needs rephrasing. 

Response: Manuscript revised, lines 485-486

Line 454: “When individuals are ready to commence with camera activation” how is this decided and how it varies? 

Response: Manuscript revised see line 547

Line 460-462: “(i.e. depending on the device there are between 3 to 4 different image processing modes e.g. white-on-black, inverse (black-on-white), edge detection, and motion detection).” Can a figure representing all these different modes be added? 

Response: While we agree such a figure would be interesting, neither author was able to observe all four modes and what they looked like on the interface and thus their appearance would be speculation.

Line 482: “resulting in the participants must make constant scanning” resulting in participants having to make 

Response: Manuscript revised see line 527

Line 488: “they zone in on an object” I believe you mean zoom in. 

Response: Manuscript revised see line 533

Line 493: “in this way it requires that they rethink the concept of “seeing.”” It requires them rethinking the concept of “seeing”. 

Response: Manuscript revised see line 539

Line 525: “He says while he passes the again.” He passes the sheet in front Isabelle again. 

Response: Manuscript revised see line 567

It is very useful to have a clear description of all the phases involved in the process required to deliver and use a retinal implant, however the data and subjective reports are scarce given the 16 participants tested, it would be good to have more examples of participants reports throughout the manuscript and a table with examples on a specific theme from all participants to better understand the subjective variability and experience with the implants, also it would be useful to report, the age of the participant what implant the participant was using as two are examined here when reporting quotations [e.g. quote, Participant 1, female, 56 years, ARGUS II] at the moment the manuscript reads more as a description of the retina implant process than of the subjective experience of the participants which was the main aim of the study.

Response: All sixteen participants’ data are reflected in our major findings. There were nine participants from whom we quote directly; the other seven were not rejected or discarded - we simply did not quote from them directly. This is the standard for ethnographic research. 

The subjective experience of the participants cannot be taken apart from the process the participants go through in receiving and learning to use the device. For instance, the assumptions that have gone into structuring the rehabilitation process and the ways in which it is implemented play a significant role in determining what the participant perceives (as we say in the manuscript - “the process of teaching an individual to use the device is so specific to the dynamic between participant and rehabilitation specialist that it can be considered to be “co-constructed” within their interactions”). We are also interested in their more general subjective experience of receiving one of these devices, which includes their cognitive fatigue and disappointment (for example). The context of getting learning and living with the device is essential in situating these findings. 

We believe tables of the kind you mention are more suitable for studies that are able to control for task and stimulation parameters - common to psychophysics - see Fornos et al 2012 or Chen et al, 2009. Here, where our sampling was opportunistic (as is common for ethnographic studies), and standardization unfeasible, a table makes less sense.

Finally, because there are so few participants, including the participants’ age and device type risks identifying (particularly by the manufacturers, who many of the participants explicitly requested to remain anonymous to (as a condition of being candid). As mentioned above and in the cover letter, this is why we cannot make our dataset publicly available.

Fornos, AP et al. “Temporal Properties of Visual Perception on Electrical Stimulation of the Retina. Invest.” Ophthalmol. Vis. Sci. 2012;53(6):2720-2731.

Chen, SC et al. Simulating prothetic visin: I. Visual models of phosphenes. Vision Research. 2016;49(12):1493-1506 

Line 614-615: please use participants rather than ‘them’ twice in a sentence, e.g “The vision therapist walks along beside them, tracking the translation of the video camera image into electrode activation on a laptop they tow alongside them on a wheeled walker.” The vision therapist walks along beside the participant, ….them on a wheeled walker. 

Response: Manuscript revised see lines 661

Line 694: “This accords with published literature” better “this is in agreement or in line with”, also some more references are needed in these sentence, what published literature? Is just one study? [23]? Explain how is in agreement with this study, what did they show? 

Response: Manuscript cites Stronks et al 2014 study which is a comprehensive review of the clinical literature and reflects the conclusion that we are citing

Line 794: “lest it endanger” because it could endanger. 

Response: Manuscript revised see line 845

Line 823-827: “While “visual” in terms of being similar to what participants remember of the experience of certain kinds of light, as well as by offering the possibility of being able to understand features of the environmental surround at a distance, artificial vision was described as both qualitatively and functionally different than the “natural” vision the participants remember.” Very long and difficult to follow sentence consider splitting it into two sentences and rephrase it. 

Response: Manuscript revised, see lines 878-880

Line 829: “and more a substitution; that is, as offering an entirely different or novel sensory tool” I agree but then there are lots of sensory substitution devices (e.g. The vOICe) that are less invasive and less difficult to use and are free, so why going through all this process? A discussion of this point would be important as these implants are as you mention cognitively heavy and at least similar (if not at times better) goals can be achieved with sensory substitution devices relying on available and intact senses such as sound and touch rather than lost vision. 

Response: We whole-heartedly agree that this is the kind of question this analysis leads to. Our goal here, however, is not to take a stance or position on whether these devices are worthwhile. Our intention is to present the data and leave the discussion of the kind you mention to the researchers, industry and most importantly, potential participants. 

Line 869: “The more early” the earlier. 

Response: Manuscript revised see line 924

Line 876: “we found that that the participant’s” delete one “that”. 

Response: Manuscript revised see line 931

In the discussion it would be good to underline how the use of these implants is discontinued by the far majority of patients and whether anything emerged in the interviews and data that could be done to improve this outcome. 

Response: We have revised the results to mention the report of one researcher that it has been their experience that the use of these implants is discontinued by a majority of recipients (see lines 782-785). Suggestions for how these findings might be used to inform design and implementation discussed in lines 884-951.

---

## [Decision Letter · Decision Letter 1]

26 Nov 2020

PONE-D-20-02730R1

What do blind people “see” with retinal prostheses? Observations and qualitative reports of epiretinal implant users

PLOS ONE

Dear Dr. Erickson-Davis,

Thank you for submitting your manuscript to PLOS ONE. After careful consideration, we feel that it has merit but does not fully meet PLOS ONE’s publication criteria as it currently stands. Therefore, we invite you to submit a revised version of the manuscript that addresses the points raised during the review process.

Only minor changes are required, which I wanted you to have the opportunity to address quickly. The manuscript will not have to go back to review, provided you only make the changes suggested by the reviewers. 

We look forward to receiving your revised manuscript.

Kind regards,

Nicholas Seow Chiang Price, Ph.D.

Academic Editor

PLOS ONE

Reviewers' comments:

Reviewer's Responses to Questions

**Comments to the Author**

1. If the authors have adequately addressed your comments raised in a previous round of review and you feel that this manuscript is now acceptable for publication, you may indicate that here to bypass the “Comments to the Author” section, enter your conflict of interest statement in the “Confidential to Editor” section, and submit your "Accept" recommendation.

Reviewer #1: All comments have been addressed

Reviewer #2: All comments have been addressed

2. Is the manuscript technically sound, and do the data support the conclusions?

Reviewer #1: Yes

Reviewer #2: Yes

3. Has the statistical analysis been performed appropriately and rigorously? 

Reviewer #1: N/A

Reviewer #2: Yes

4. Have the authors made all data underlying the findings in their manuscript fully available?

Reviewer #1: Yes

Reviewer #2: No

5. Is the manuscript presented in an intelligible fashion and written in standard English?

Reviewer #1: Yes

Reviewer #2: Yes

6. Review Comments to the Author

Reviewer #1: Excellent, substantive revisions. This is an important piece of work, and I'm excited to see more qualitative discussion of the experience of prosthetic vision in the field.

I'm happy with the paper as-is, but the authors could continue to check the added text for typos/grammatical errors.

Reviewer #2: I have very few minor comments

There seems to be a comment in the supplemental material that the authors may want to delete

In the abstract ‘scores of them closely” is unclear, scores of what?

From page 16. P1 P2 etc…shouldn’t they be R1, R2 ect…given the use of Recipient 1, 2 etc…?

7. PLOS authors have the option to publish the peer review history of their article (what does this mean?). If published, this will include your full peer review and any attached files.

Reviewer #1: No

Reviewer #2: No

---

## [Author Response · Author response to Decision Letter 1]

28 Nov 2020

Reviewer #1: Excellent, substantive revisions. This is an important piece of work, and I'm excited to see more qualitative discussion of the experience of prosthetic vision in the field.

I'm happy with the paper as-is, but the authors could continue to check the added text for typos/grammatical errors.

Reviewer #2: I have very few minor comments

There seems to be a comment in the supplemental material that the authors may want to delete

Response: Thanks. Comment is taken out

In the abstract ‘scores of them closely” is unclear, scores of what?

Response: It refers to the “hundreds of individuals implanted” that immediately precedes the statement. Scores – meaning dozens – of the individuals implanted with retinal implants have been followed in research trials. We have checked with editor who has confirmed this is grammatically correct.

From page 16. P1 P2 etc…shouldn’t they be R1, R2 ect…given the use of Recipient 1, 2 etc…?

Response: Thanks for catching this. P[#} has been changed to R[#] throughout manuscript.

---

## [Editor Report · Decision Letter 2]

1 Dec 2020

What do blind people “see” with retinal prostheses? Observations and qualitative reports of epiretinal implant users

PONE-D-20-02730R2

Dear Dr. Erickson-Davis,

We’re pleased to inform you that your manuscript has been judged scientifically suitable for publication and will be formally accepted for publication once it meets all outstanding technical requirements.

Kind regards,

Nicholas Seow Chiang Price, Ph.D.

Academic Editor

PLOS ONE
---

## [Editor Report · Acceptance letter]

18 Jan 2021

PONE-D-20-02730R2 

What do blind people “see” with retinal prostheses? Observations and qualitative reports of epiretinal implant users 

Dear Dr. Erickson-Davis:

I'm pleased to inform you that your manuscript has been deemed suitable for publication in PLOS ONE. Congratulations! Your manuscript is now with our production department. 

Kind regards, 

on behalf of

Dr. Nicholas Seow Chiang Price 

Academic Editor

PLOS ONE